# Photothermal Therapy with HER2-Targeted Silver Nanoparticles Leading to Cancer Remission

**DOI:** 10.3390/pharmaceutics14051013

**Published:** 2022-05-08

**Authors:** Victoria O. Shipunova, Mariia M. Belova, Polina A. Kotelnikova, Olga N. Shilova, Aziz B. Mirkasymov, Natalia V. Danilova, Elena N. Komedchikova, Rachela Popovtzer, Sergey M. Deyev, Maxim P. Nikitin

**Affiliations:** 1Department of Nanobiomedicine, Sirius University of Science and Technology, 1 Olympic Ave., 354340 Sochi, Russia; belova.mm@phystech.edu (M.M.B.); max.nikitin@phystech.edu (M.P.N.); 2Shemyakin-Ovchinnikov Institute of Bioorganic Chemistry, Russian Academy of Sciences, 16/10 Miklukho-Maklaya St., 117997 Moscow, Russia; kotelnikova@phystech.edu (P.A.K.); olchernykh@yandex.ru (O.N.S.); zika131@mail.ru (A.B.M.); deyev@ibch.ru (S.M.D.); 3Moscow Institute of Physics and Technology, 9 Institutskiy Per., 141701 Dolgoprudny, Russia; lena-kom08@rambler.ru; 4Faculty of Medicine, Lomonosov Moscow State University, 27/1 Lomonosovsky Ave., 119192 Moscow, Russia; ndanilova@mc.msu.ru; 5Faculty of Engineering, Institute of Nanotechnology & Advanced Materials, Bar-Ilan University, Ramat Gan 5290002, Israel; rachela.popovtzer@biu.ac.il

**Keywords:** silver nanoparticles, photothermal therapy, local hyperthermia, targeted delivery, HER2, affibody, green synthesis, Ag

## Abstract

Nanoparticles exhibiting the localized surface plasmon resonance (LSPR) phenomenon are promising tools for diagnostics and cancer treatment. Among widely used metal nanoparticles, silver nanoparticles (Ag NPs) possess the strongest light scattering and surface plasmon strength. However, the therapeutic potential of Ag NPs has until now been underestimated. Here we show targeted photothermal therapy of solid tumors with 35 nm HER2-targeted Ag NPs, which were produced by the green synthesis using an aqueous extract of *Lavandula angustifolia* Mill. Light irradiation tests demonstrated effective hyperthermic properties of these NPs, namely heating by 10 °C in 10 min. To mediate targeted cancer therapy, Ag NPs were conjugated to the scaffold polypeptide, affibody Z_HER2:342_, which recognizes a clinically relevant oncomarker HER2. The conjugation was mediated by the PEG linker to obtain Ag-PEG-HER2 nanoparticles. Flow cytometry tests showed that Ag-PEG-HER2 particles successfully bind to HER2-overexpressing cells with a specificity comparable to that of full-size anti-HER2 IgGs. A confocal microscopy study showed efficient internalization of Ag-PEG-HER2 into cells in less than 2 h of incubation. Cytotoxicity assays demonstrated effective cell death upon exposure to Ag-PEG-HER2 and irradiation, caused by the production of reactive oxygen species. Xenograft tumor therapy with Ag-PEG-HER2 particles in vivo resulted in full primary tumor regression and the prevention of metastatic spread. Thus, for the first time, we have shown that HER2-directed plasmonic Ag nanoparticles are effective sensitizers for targeted photothermal oncotherapy.

## 1. Introduction

Over the past few decades, the rapid development of nanotechnology and protein engineering has led to the creation of new schemes of diagnostics and treatment of socially significant diseases, including cancer [1,2,3,4]. Nanostructures, possessing unique properties such as fluorescence or high drug encapsulation capacity, are considered to be the most effective platform for the creation of smart methods of personalized diagnostics and cancer therapy [5,6,7,8].

During the transition to nanoscale dimensions, bulk materials acquire unusual physical and chemical properties thus making possible the design of multifunctional tools for biomedicine. One such interesting property is the effect of localized surface plasmon resonance (LSPR) of metal nanoparticles (NPs). When a nanoparticle is irradiated with incident light and the LSPR conditions are met, the intensity of reflected light decreases sharply due to the transition of the energy of the incident electromagnetic wave into the energy of a plasmon. These properties of nanostructures are effectively used in the development of diagnostic and therapeutic systems [9,10,11]. The absorbed energy can be converted into thermal energy, thereby realizing the hyperthermic properties of the LSPR sample, which can be used for localized hyperthermia of cancer cells [12].

To date, a wide variety of LSPR nanoparticles, such as silver, aluminum, copper, palladium, and platinum, have been studied, and their effective use has been shown not only for biomedical applications but also in biophysics and biosensorics due to their biocompatibility, inertness, and ease of chemical modification through sulfhydryl chemistry. Fundamental research directed toward the development of anticancer drugs based on LSPR particles is mainly focused on gold nanoparticles. However, among various metal nanoparticles, silver nanoparticles possess the most effective light trapping potential due to their strong light scattering and surface plasmon strength [13]. However, the potential of silver nanoparticles in biomedical applications, namely for cancer therapy in vivo, was underestimated until today.

The research of Ag particles was predominantly focused on the in vitro evaluation of their cytotoxic properties, e.g., studying the BSA-coated silver particles for inhibiting HUVEC cell proliferation [14], chitosan-coated particles for nonsmall lung cancer cell growth inhibition [15], or PVP-stabilized Ag NPs for colon tumor cell growth inhibition [16]. A limited number of in vivo studies have been focused on investigating the bioimaging potential of Ag nanoparticles [17,18,19] or studying the accumulation of nontargeted Ag particles in the tumor due to enhanced permeability and retention effect [20].

The development of targeted drug delivery methods significantly enhanced the performance of nanoparticle-based anticancer drugs, including metallic nanoparticles. Monoclonal IgG antibodies and their derivatives were traditionally used for diagnostics and targeted drug delivery; however, they have a whole range of undesirable side effects [21,22,23]. Synthetic nonimmunoglobulin-recognizing scaffold proteins, e.g., affibodies, DARPins, or ADAPTs, appear to be much more effective tools for targeting nanostructures to cancer cells than traditional IgGs due to the ease of large-scale biotechnological production in bacteria, high stability in severe conditions, and the absence of immunogenicity in vivo [19,21,22,24,25,26,27,28]. In particular, one of the most promising scaffolds is the affibody Z_HER2:342_, which highly selectively recognizes the clinically relevant oncomarker HER2 and is already undergoing clinical trials [29].

Here we introduce highly specific nanoagents for cancer therapy, which is a combination of plasmonic Ag NPs and the anti-HER2 affibody Z_HER2:342_. The Ag NPs were synthesized via the "green" synthesis using the aqueous extract of *Lavandula angustifolia* Mill. grown in vitro using standard protocols. Such a synthesis method allowed us to combine the therapeutic functions of Ag^+^ ions and secondary metabolites of the medicinal plant lavender, and the modification with the targeted scaffold makes it possible to integrate diagnostic functions of affibody targeting and fluorescence imaging.

The as-synthesized 30 nm Ag NPs possess pronounced hyperthermic properties under irradiation with an external light source, which was used for successful photothermal therapy (PTT) of HER2-overexpressing tumors with affibody-decorated Ag NPs. We showed that such targeted Ag NPs in combination with light irradiation lead to complete cancer remission. Thus, the affibody-equipped targeted silver nanoparticles should be considered as promising agents for the HER2-overexpressing tumor treatment using the local hyperthermia phenomenon.

## 2. Materials and Methods

### 2.1. Experimental Design

The design of the research is schematically presented in Figure 1, which includes two consecutive parts of the study:(i)Synthesis, characterization, and modification of silver nanoparticles. First, the optimal conditions for the NP synthesis were found to obtain stable particles in the sub100 nm size range. Then, physicochemical, optical, and photothermal properties of NPs were studied. To introduce cancer cell-targeting capabilities, particles were equipped with anti-HER2 scaffold protein, namely affibody Z_HER2:342_, through intermediate PEG modification.(ii)Study of nanoparticle–cell interaction, cytotoxic properties in vitro, and antitumor activity in vivo. First, the processes of interaction of NPs with HER2-overexpressing cancer cells were studied, namely the NP internalization rate and the cytotoxic properties. Next, the light-induced hyperthermia of cancer cells in vitro was demonstrated. Last, the anticancer activity of NPs in vivo using the xenograft HER2-overexpressing tumors was studied. Cancer remission was shown when mice were treated with injections of targeted Ag NPs followed by light irradiation.

### 2.2. Materials

Sigma-Aldrich, Darmstadt, Germany: sodium phosphate dibasic, dehydrate, kanamycin, Tris base, N-(3-Dimethylaminopropyl)-N′-ethylcarbodiimide hydrochloride (EDC), sodium phosphate monobasic, monohydrate, and dimethyl sulfoxide; PanReac, Barcelona, Spain: glycerol, triptone, acrylamide, imidazole, and silver nitrate; Serva, Germany: ammonium persulfate, SDS, 3-(4,5-Dimethyl-2-thiazolyl)- 2,5-diphenyl- 2H-tetrazolium·bromide, and MTT; Molekula Ltd., Darlington, UK: TEMED, tricine, and DTT; Gerbu, Germany: yeast extract for microbiology; AppliChem, Darmstadt, Germany: sodium chloride; Thermo Fisher Scientific, Waltham, MA, USA: Coomassie blue G-250, BCA protein assay, sulfo-NHS, CM-H_2_DCFDA, Hoechst33342, DyLight650-NHS ester, and FITC; Paneco, Moscow, Russia: Versene solution, L-glutamine, penicillin-streptomycin, and bovine serum albumin; Roche, Mannheim, Germany: Herceptin; HyClone, Logan, UT, USA: DMEM medium and fetal bovine serum; Dia-M, Moscow, Russia: N,N′-Methylenebis(acrylamide); Bioveta, Ivanovice na Hané, Czech Republic: Rometar; Virbac, Carros, France: Zoletil; Corning Life Sciences, New York, NY, USA: Matrigel; Olvex Diagnosticum, Russia: AST-RF-OLVEX kit, ALT-RF-OLVEX kit, LDH-OLVEX kit, UREA-2-OLVEX kit, and Alkaline Phosphatase-2-OLVEX kit; Promega, USA: furimazine; LenReaktiv, St. Petersburg, Russia: crystal violet dye; Biopharma PEG Scientific Inc., Watertown, MA, USA: mPEG-silane-COOH 5 kDa.

### 2.3. Nanoparticle Synthesis

Silver NPs were prepared using the “green” synthesis technique by mixing 50 μL of AgNO_3_ solution at 1 g/L in water with the 50 μL of *Lavandula angustifolia* Mill. plant extract in the concentration range starting from 30% with 2-fold serial dilution down to 0.5%. *Lavandula angustifolia* Mill. was cultured in vitro, and the extract was obtained as described by us previously [30]. During the particle synthesis, absorption spectra at 350–800 nm were measured at different time points (30, 60, 150, and 240 min) using an Infinite M1000 Pro microplate reader (Tecan, Grödig, Austria). The efficiency of NP synthesis was evaluated by the LSPR peak intensity. The LSPR peak is considered to be a qualitative criterion for the presence of metallic NPs in a system [31].

### 2.4. Scanning Electron Microscopy

The morphology of the synthesized Ag NPs was investigated using the MAIA3 Tescan (Tescan, Brno-Kohoutovice, Czech Republic) scanning electron microscope at an accelerating voltage of 10 kV. Samples of nanoparticles at 5 µg/mL in water were air-dried on a silicon wafer on carbon film and analyzed immediately. SEM images were evaluated using ImageJ software to obtain a particle size distribution.

### 2.5. Hyperthermic Properties Study

The 5 × 5 cm LED matrix was used to irradiate NPs at a wavelength of 465 nm and a power of 95 mW/cm^2^.

To study the temperature change dynamics of Ag NP suspensions, samples at different concentrations (0.03–2.2 g/L) in H_2_O were placed under LEDs and irradiated for 25 min. Temperature data were recorded every 30 s using an ETS-D5 electronic contact thermometer (IKA, Heidolph, Schwabach, Germany). The sample with H_2_O was used as the control sample.

### 2.6. Affibody Z_HER2:342_ Isolation and Purification

*E. coli* (strain BL21(DE3)), transformed with pET39-Z342 gene [32], was grown in autoinduction ZYM-5052 medium containing 100 μg/mL kanamycin at 25 °C for 24 h at 25 °C. The cells were harvested by centrifugation at 10,000 g at 4 °C for 15 min and resuspended in lysis buffer (20 mM Na-Pi, 300 mM NaCl, pH 7.5, and 50 μg/mL lysozyme). The suspension was diluted with distilled water at 1:1 and incubated at room temperature for 30 min. Cells were broken on ice with the Vibra Cell ultrasonic liquid processor VCX130 (Sonics & Materials, Inc., Newtown, CT, USA). The debris was centrifuged at 30,000× *g* at 4 °C for 120 min. After the addition of imidazole (to obtain a final concentration of 25 mM), the supernatant was filtered through a 0.22 μm membrane and applied onto a HisTrap HP, 1 mL column (GE Healthcare) equilibrated with 20 mM sodium phosphate buffer (pH 7.5), 300 mM NaCl, and 25 mM imidazole. The bound proteins were eluted with imidazole step gradient (50, 100, 150, 200, 250, and 500 mM).

The fractions were analyzed by SDS-PAGE (10%) using the Tris-tricine discontin-uous electrophoresis system consisting of separating gel (9.8% acrylamide/0.26% bisacrylamide, 13% glycerol, 0.05% ammonium persulfate, 0.001% TEMED, 0.1% SDS, and 1M Tris⋅Cl, pH 8.45), stacking gel (3.9% acrylamide/0.1% bisacrylamide, 0.05% ammonium persulfate, 0.001% TEMED, 0.075% SDS, and 0.75 M Tris⋅Cl, pH 8.45), anode buffer (0.2 M Tris base, pH 8.9), cathode buffer (0.1 M Tris base and 0.1 M tricine), and sample buffer (24% glycerol, 8% SDS, 0.02% Coomassie blue G-250, 0.2 M DTT, and 0.1 M Tris⋅Cl, pH 6.8).

Protein concentration was determined by BCA protein assay (Thermo Fisher Scientific, New York, NY, USA).

### 2.7. Nanoparticle Modification: BSA Stabilization, PEG Modification, and Affibody Conjugation

For the cell toxicity assays, Ag NPs were stabilized with bovine serum albumin (BSA). The modification was carried out through the formation of a noncovalent bond between the—SH groups of the protein and the particle surface. To reduce disulfide bonds, BSA at 1 g/L was mixed with dithiothreitol at 5 g/L at the 27:1 ratio in dH_2_O. After 5 min of incubation at room temperature, the protein was mixed with NPs at 1.8 g/L in a 1.75:1 v/v ratio; the suspension was gently mixed, briefly exposed to ultrasound, and incubated for 8 h at room temperature. Next, phosphate-buffered saline (PBS) was added to the mixture of Ag NPs; Ag-BSA particles were washed twice from unbound protein by centrifugation for 10 min at 15,000× *g*, and, finally, Ag-BSA particles were resuspended in PBS.

Silver NPs directed toward HER2 were obtained through intermediate PEGylation and subsequent carbodiimide conjugation with affibody Z_HER2:342_. Methoxyl silane functionalized polyethylene glycol (mPEG-silane-COOH) at 4 g/L in 96% ethanol was mixed with Ag NPs at 1.8 g/L in H_2_O in a 4:1 ratio; the mixture was incubated for 8 h at room temperature, and then PEG-modified Ag NPs were washed twice from unbound PEG derivatives by centrifugation for 5 min at 15,000× *g* in H_2_O.

For the carbodiimide conjugation, Ag-PEG NPs were mixed with 1-ethyl-3-(3-dimethyl aminopropyl)carbodiimide (EDC) at 15.4 g/L in 0.1 M MES (2-(N-morpholino) ethanesulfonic acid), pH 5.0 buffer, and N-hydroxysuccinimide (NHS) at 50 g/L in dimethyl sulfoxide (DMSO) at 2.5:6.5:1 ratio. A total of 100 µL of this mixture was incubated for 20 min at room temperature and then centrifuged for 3 min at 5000× *g*. Next, 100 μL of affibody Z_HER2:342_ at 0.2 g/L in borate buffer (0.4 M H_3_BO_3_, 70 mM Na_2_B_4_O_7_, pH 8.0) was added to Ag-PEG particles. The mixture was incubated for 8 h at room temperature; then 200 μL of PBS with 1% BSA was added, and NPs were washed with triple centrifugation for 3 min at 5000× *g* and resuspended in PBS 1% BSA to obtain Ag-PEG-Z_HER2:342_ nanoparticles. The efficiency of nanoparticle modification was indirectly confirmed by the measurement of their hydrodynamic size. Particle size was determined by dynamic light scattering on a Zetasizer Nano ZS analyzer (Malvern Instruments, Ltd. Worcestershire, UK).

### 2.8. Cell Culture

Human breast adenocarcinoma BT-474 (ATCC HTB-20), SK-BR-3 (ATCC HTB-30), and MCF-7 (ATCC HTB-22) cells; human ovarian adenocarcinoma SK-OV-3-1ip cells (Shemyakin-Ovchinnikov Institute RAS, Molecular Immunology Laboratory collection, [33]); human lung carcinoma A549 (ATCC CCL-185) cells; and Chinese hamster ovary CHO (Shemyakin-Ovchinnikov Institute RAS, Molecular Immunology Laboratory collection) cells were cultured in DMEM medium (HyClone, Logan, UT, USA) supplemented with 10% fetal bovine serum (HyClone, Logan, UT, USA), penicillin/streptomycin (PanEko, Moscow, Russia), and 2 mM L-glutamine (PanEko, Moscow, Russia). Cells were incubated under a humidified atmosphere with 5% CO_2_ at 37 °C. BT/NanoLuc cells were obtained by us previously and used without any modifications [32].

### 2.9. FITC and DyLight650 Conjugation

Affibody Z_HER2:342_ was conjugated to FITC as described by us previously [32].

For the flow cytometry tests, Trastuzumab was conjugated to DyLight650 as follows: A total of 100 µg of Trastuzumab in 90 µL of PBS was rapidly mixed with 10 µL of DyLight650 NHS ester in DMSO at a concentration of 0.54 g/L. The protein was incubated for 8 h at RT and purified from unreacted molecules using Zeba Spin Desalting Columns according to the manufacturer′s recommendations.

A total of 100 µg of Ag-PEG-HER2 nanoparticles in 100 µL of PBS was rapidly mixed with 10 µL of DyLight650 NHS ester in DMSO at concentrations of 0.1 g/L to obtain Ag-PEG-HER2*DyLight650. Particles were incubated for 3 h at RT and purified from unreacted molecules with triple washing by centrifugation in PBS with 1% BSA.

### 2.10. Confocal Microscopy

To visualize Z_HER2:342_-FITC binding to cells by confocal laser scanning microscopy, HER2-overexpressing SKOV3-1ip and HER2-negative CHO cells were incubated with 2 µg/mL of Z_HER2:342_-FITC and Hoechst33342 (1 µg/mL) on ice for 30 min in PBS with 1% BSA, washed from unbound molecules, followed by confocal laser scanning microscopy imaging using an LSM 980 (Zeiss) confocal microscope at the following conditions: excitation 488 nm, emission 492–550 nm for FITC detection and excitation 405 nm, and emission 410–520 nm for Hoechst33342 detection.

### 2.11. Flow Cytometry

To determine Z_HER2:342_-FITC cell-binding efficiency, the harvested SKOV3-1ip and CHO cells were washed with PBS, resuspended in 300 µL of PBS with 1% BSA at a concentration of 10^6^ cells/mL, labeled with Z_HER2:342_-FITC at 2 µg/mL, washed from unbound molecules, and analyzed using Novocyte 3000 VYB flow cytometer (ACEA Biosciences, San Diego, CA, USA) in BL1 channel (excitation laser 488 nm, emission filter 530/30 nm).

The interaction of modified Ag-PEG-HER2 with cells was studied as follows: 2.5 × 10^5^ cells in 300 μL of PBS 1% BSA was mixed with Ag-PEG-HER2 particles and incubated for 15 min. The cells were then washed from unbound particles by centrifugation for 3 min at 100× *g*, resuspended in 300 μL PBS with 1% BSA, and analyzed with Novocyte 3000 VYB flow cytometer (ACEA Biosciences, USA) using the side scattering parameter SSC-A for particle detection.

To determine Trastazumab-DyLight650 and Ag-PEG-HER2*DyLight650, SKOV3-1ip, BT 474, SK-BR-3, A549, MCF-7, and CHO cell-binding efficiency, cells were labeled with Trastazumab-DyLight650 at 2 µg/mL and Ag-PEG-HER2*DyLight650 at 4 µg/mL, washed, and analyzed using Accuri C6 (BD) flow cytometer in FL4 channel (excitation laser 640 nm and emission filter 675/25 nm).

### 2.12. Viability Assay

The cytotoxicity of NPs was investigated using the MTT test. CHO and SKOV3-1ip cells were plated onto a 96-well plate at 2.5 × 10^3^ cells per well in 100 μL of DMEM medium with 10% FBS. After overnight culturing, 100 μL of either DMEM 10% FBS or DMEM 10% FBS with nanoparticles at various concentrations was added to the cells. The cells were incubated for 48 h in a 5% CO_2_ atmosphere at 37 °C. Then the medium was removed, and the cells were washed once with the medium. Next, 100 μL of 0.5 g/L MTT solution (tetrazolium dye, 3-(4,5-dimethyl-2-thiazolyl)-2,5-diphenyl-2H-tetrazolium bromide) in serum-free DMEM medium was added to the wells and incubated for 1 h in a 5% CO_2_ atmosphere at 37 °C. Next, the MTT solution was removed, and 100 μL of DMSO (dimethyl sulfoxide) was added to the wells; the plate was shaken until the formazan crystals were completely dissolved. The optical density of wells was measured using the Infinite M100 Pro microplate reader (Tecan, Grödig, Austria) at a wavelength of λ = 570 nm.

For the photothermal-induced cytotoxicity study, after the addition of Ag NPs to cells, incubation, and washing from nonbound nanoparticles, the plate was placed under the 5 × 5 cm LED matrix (465 nm, power of 95 mW/cm^2^) and irradiated for 20 min followed by the cultivation and cytotoxicity study as described above.

### 2.13. Clonogenic Assay

The influence of Ag-PEG-Z_HER2:342_ on the proliferative activity of SKOV3-1ip cells was carried out with the clonogenic analysis. Cells were placed onto a 12-well plate at 10^3^ cells per well in 1000 μL of DMEM medium with 10% FBS. After culturing the cells at 37 °C in a CO_2_ incubator overnight, the test samples were sterilely added to the cells at various concentrations (2.2–7.6 μg per well). Immediately after adding NPs and after 120 min of incubation, 12-well plates with cells were placed under LEDs and irradiated for 20 min. For control experiments, 12-well plates with cells and nanoparticles were used without irradiation. The cells were incubated for 14 days at 37 °C in a CO_2_ incubator. Then, the medium was removed, and 300 μL of a 1:1 mixture of PBS and methanol was added to the wells and incubated for 2 min. After the incubation, the solution was removed, and 300 μL of methanol was added and incubated for 10 min. Next, methanol was removed, and the cells were washed once with distilled water and added to the wells with 300 μL of 1% crystal violet dye solution in water and incubated for 10 min. Next, the solution was removed. The proliferative activity of cells was assessed by the number of colonies consisting of 50 or more cells. Unirradiated cells cultured without the addition of Ag-PEG-Z_HER2:342_ NPs served as control. The images of cell colonies were taken during clonogenic analysis using a Stemi DV4 stereomicroscope (Carl Zeiss, Jena, Germany) in transmitted light at 4-fold magnification.

### 2.14. ROS Generation Study

The ROS generation in cell samples incubated with modified NPs was studied using the flow cytometry assay with the general oxidative stress indicator CM-H_2_DCFDA. 4.5 × 10^5^ cells were incubated with modified NPs in DMEM with 10% FBS for 30 min, washed from nonbound NPs with centrifugation for 5 min at 100× *g*, and resuspended in 500 μL of phenol red-free DMEM medium with 10% FBS. Then samples were irradiated with LED matrix at a power of 95 mW/cm^2^ at 10 °C for 0, 2, and 5 min. Cells with nanoparticles without irradiation were also investigated under the same conditions. Then CM-H_2_DCFDA, the general oxidative stress indicator, was added according to the manufacturer′s recommendations. The samples were incubated for 1.5 h in the dark at 37 °C and then placed on ice and analyzed immediately with the Accuri C6 flow cytometer (BD Biosciences, San Jose, CA, USA) in the FL1 channel (excitation laser 488 nm and emission filter 533/30 nm). Cells without NPs (autofluorescence) served as control.

### 2.15. Tumor-Bearing Mice

The antitumor efficacy of the synthesized targeted NPs in vivo was investigated using a xenograft tumor model in BALB/c Nu/Nu mice. Female BALB/c Nu/Nu mice of 20–24 g weight were used in the experiments. All procedures with animals were approved by the IBCh RAS Institutional Animal Care and Use Committee.

### 2.16. In Vivo Therapy

The animals were anesthetized with a mixture of Zoletil (Virbac, Carros, France) and Rometar (Bioveta, Ivanovice na Hané, Czech Republic) at a dose of 25/5 mg/kg. Mice were s.c. injected with 2 × 10^6^ BT/NanoLuc cells in 100 μL of 30% Matrigel in the full culture medium. Next, mice were randomly divided into three groups that received: (i) intratumoral injections of 320 μg of NPs modified with affibody Z_HER2:342_ (Ag-PEG-HER2), (ii) intratumoral injections of Ag-PEG-HER2 and external light irradiation, and (iii) intratumoral injections of 100 μL PBS. All injections were performed 3 times on days 9, 12, and 15 after the tumor inoculation. The irradiation was carried out 1 h after Ag-PEG-HER2 injection for 20 min; such an irradiation scheme was selected based on the data obtained in the in vitro tests. Tumor growth was measured every three days for 90 days. The tumor size was measured with a caliper using the formula V = width^2^ × length/2. At the end of the experiment, mice were euthanized by cervical dislocation, and the primary tumor was visualized ex vivo or evaluated with immunohistochemistry.

### 2.17. In Vivo Toxicity

To evaluate toxicity in vivo, 200 μL of mice blood samples was collected without heparin. The following biochemical parameters were determined in mouse serum: aspartate aminotransferase (AST) was determined using a commercial AST-RF-OLVEX kit (#002.001, Olvex Diagnosticum, St. Petersburg, Russia); alanine aminotransferase (ALT) was determined using a commercial ALT-RF-OLVEX kit (#001.001, Olvex Diagnosticum, Russia); lactate dehydrogenase (LDH) was determined using a commercial LDH-OLVEX kit (#023.001, Olvex Diagnosticum, Russia); creatinine (CREA) was determined using a commercial CREATININE-D-OLVEX kit (#004.002, Olvex Diagnosticum, Russia); UREA was determined using a commercial UREA-2-OLVEX kit (#008.002, Olvex Diagnosticum, Russia); alkaline phosphatase (ALP) was determined using a commercial ALKALINE PHOSPHATASE-2-OLVEX kit (#009.002, Olvex Diagnosticum, Russia).

### 2.18. In Vivo Bioimaging

Bioimaging in vivo was performed by visualizing the bioluminescence of luciferase NanoLuc in BT/NanoLuc cells in vivo after the intraperitoneal injection of the NanoLuc substrate—furimazine (3 μg in 150 μL of PBS) using the IVIS Spectrum CT (PerkinElmer, Waltham, MA, USA) bioimaging system.

## 3. Results

### 3.1. Synthesis and Characterization of Silver Nanoparticles for HER2-Positive Cancer PTT

Silver NPs for targeted cancer PTT were synthesized via the reduction of silver nitrate AgNO_3_ by secondary metabolites of *Lavandula angustifolia* Mill. aqueous extract. *Lavandula angustifolia* Mill. is a widely used plant in biotechnology that can be grown in a large quantities under in vitro conditions to obtain a batch-to-batch reproducible aqueous extract [34].

The synthesis process represents a one-pot method with an operating time of *ca.* 3 min and without the use of any harmful compounds such as sodium borohydride or CTAB. To obtain nanoparticles, the AgNO_3_ solution was gently mixed with the aqueous extract of *Lavandula angustifolia* Mill. at different concentrations, and the solution was further incubated without any interventions. Spectrophotometric monitoring of NP synthesis dynamics indicated a monotonic increase reaching a plateau relationship between the absorption of the Ag NP sample at a wavelength corresponding to the LSPR peak and the concentration of the extracts used, as well as the incubation time of the silver salt with the extract (Figure 2a). The highest concentration of Ag NP sample was obtained during the synthesis with 30% extract for 240 min and corresponded to 0.82 r.u. of the NP sample at the wavelength of the LSPR peak. Under such conditions, the sample adsorption exceeded similar value at the minimum 0.5% extract concentration 205 times. However, to maintain an excess of silver salt in the solution to ensure batch-to-batch reproducibility, we used the intermediate extract concentration equal to 7.5%, which led to a reproducible synthesis reaction independent of the varied room temperature and other environmental laboratory conditions.

The scanning electron microscopy data show mostly rounded shapes of as-obtained NPs (Figure 2b) with an average diameter of 35.4 ± 1.6 nm according to the results of electron micrograph processing (Figure 2c). Since NP modification for the targeted delivery often implies chemical reactions in salt buffers, the aggregation and sedimentation stability of NPs are of special importance. The synthesized NPs were shown to be stable within 10 months in phosphate-buffered saline without any surface modification (no further observations were made). In contrast, plasmonic particles usually should be additionally stabilized with various compounds to exhibit colloidal stability under such conditions [35,36].

The irradiation of the Ag NP solution with a blue LED matrix led to the extensive heating of the solution thus revealing NP properties suitable for localized cancer hyperthermia. The dynamics of temperature change presented in Figure 2d show a positive correlation between the temperature change dynamics and irradiation power and time, as well as NP concentration. The temperature change is presented as the temperature of the solution with nanoparticles with the subtracted temperature of the control sample with water. The highest temperature value was achieved at an LED matrix power of 95 mW/cm^2^ after 25 min of irradiation at a NP concentration of 2.2 mg/mL and exceeded the buffer temperature change by 10 °C. Moreover, at the lowest NP concentration (0.03 mg/mL) after 25 min irradiation at the same power, the temperature change was found to be 2.9 °C, which is also suitable for successful localized hyperthermia.

### 3.2. Nanoparticle Stabilization and HER2-Overexpressing Cells Labeling with Affibody Z_HER2:342_

Since synthesized Ag NPs were not coated with any polymers and probably contained some secondary metabolites from plant extract on their surface even after washing steps, the Ag NPs possess nonspecific cytotoxicity (Figure 3a, red curves). However, this kind of cytotoxicity could be easily shielded via the coating of the particle surface with a biocompatible protein, e.g., bovine serum albumin (BSA), which is shown in Figure 3a as green curves corresponding to the cytotoxicity of Ag-BSA NPs. The results presented are the MTT assay data of the viability of two cell lines of different origins—Chinese hamster ovarian cells CHO and human ovarian cancer cells SKOV3-1ip. Indeed, the MTT assay results show that cell viability significantly increased after protein shielding of the NP surface, namely IC50 equal to 1.17 vs. 11.6 µg/mL for unmodified and BSA-coated particles, respectively, for SKOV3-1ip cells and IC50 equal to 0.55 vs. 116.4 µg/mL for CHO cells. We should note that IC50 calculated here is a relative IC50 (not absolute IC50), which means that it is the concentration required to bring the curve down to the half way point between the top and bottom plateaus of the curve. Thus, taking into account that coating nanoparticles with BSA leads to a significant increase in IC50 and the cell viability is at least 85% for SKOV3-1ip cells and 90% for CHO cells at concentrations smaller than 2 µg/mL, such particles can be considered as nontoxic under such treatment conditions [37].

Considering these particles as promising sensitizers for the targeted cancer PTT, this “shielding” protein can guide recognizing functions and target cell surface while retaining the property of cytotoxicity shielding. For this purpose, an affibody Z_HER2:342_ that specifically binds the receptor HER2 was selected and successfully used. Affibody Z_HER2:342_ is a non-IgG scaffold polypeptide possessing a high affinity constant of binding to HER2 equal to 22 pM [21,38].

Prior to the nanoparticle cell-targeting experiments, affibody Z_HER2:342_ specificity was confirmed with the confocal microscopy and flow cytometry tests. Cells possessing different expression levels of the HER2 receptor, namely HER2-overexpressing SKOV3-1ip cells and HER2-negative CHO cells, were labeled with FITC-conjugated affibody Z_HER2:342_. Data presented in Figure 3 qualitatively (Figure 3b) and quantitatively (Figure 3c) prove the Z_HER2:342_–FITC interaction specificity with HER2-positive cells.

### 3.3. Nanoparticle Modification and HER2-Overexpressing Cell Targeting with Ag-PEG-HER2 Nanoparticles

Ag NPs directed toward HER2-overexpressing cells were obtained by intermediate PEGylation and subsequent carbodiimide conjugation with affibody Z_HER2:342_ (Figure 4a). The preliminary tests showed that nanoparticle surface modifications via protein adsorption or sulfhydryl chemistry were not effective enough for further cancer cell targeting in vitro and especially for in vivo experiments. The absence of specificity of such directly modified nanoparticles is probably due to the proximity of the small molecule of affibody (8 kDa) to the nanoparticle surface. This problem, as described by us previously, can be solved by, e.g., distancing the recognizing molecule from the nanoparticle surface via a flexible linker (Shipunova et al. 2018b).

To distance the recognizing affibody Z_HER2:342_ from the Ag nanoparticle surface and avoid any sterical hindrance during the HER2 target recognition, intermediate PEGylation was performed. For this aim, we used a heterobifunctional PEG derivative, namely mPEG-silane-COOH, in ethanol solution to obtain modified Ag-PEG NPs as described schematically in Figure 4a. Covalent coupling of affibody to Ag-PEG was performed with carbodiimide chemistry using a zero-length crosslinker—EDC (1-ethyl-3-(3-dimethyl aminopropyl)carbodiimide hydrochloride) in combination with the sodium salt of hydroxysulfosuccinimide (sulfo-NHS) that increases conjugation efficiency (Figure 4a).

DLS measurements indicate the significant NP hydrodynamic size change after the PEGylation (Ag-PEG) and affibody carbodiimide conjugation (Ag-PEG-HER2). The hydrodynamic size distribution of pristine and modified particles is shown in Figure 4b. The lateral flow assay on test strips confirmed the protein conjugation to the Ag-PEG nanoparticle surface (see Appendix A).

The as-obtained Ag-PEG-HER2 NPs were then used for the targeted delivery to HER2-overexpressing SKOV3-1ip cells. The cell-targeting efficiency was estimated with the flow cytometry by calculating the change of cell population side scatter (SSC), namely ΔSSC. The SSC value corresponds to the cell surface roughness and was previously shown to be an effective parameter reflecting the quantity of cell-bound metal NPs [39,40,41] without the incorporation of any fluorescent labels into the nanoparticle structure.

The flow cytometry results show that targeted Ag-PEG-HER2 NPs specifically interact with the HER2-overexpressing cells SKOV3-1ip in a concentration-dependent mode (Figure 4d,e) and possess a negligible nonspecific interaction with control CHO cells (Figure 4c). Indeed, the binding of Ag-PEG-HER2 to SKOV3-1ip cells was 143 times higher than the binding of nontargeted Ag-BSA nanoparticles. Ag-PEG-HER2 binding to SKOV3-1ip cells was 10.6 times higher than to HER2-negative CHO cells (Figure 4c), which do not express the HER2 receptor. Within the tested concentration range, the Ag-PEG-HER2 particles significantly outperform their nontargeted counterparts Ag and Ag-PEG in terms of HER2-positive cell-binding efficiency (Figure 4e).

### 3.4. Imaging Modalities of Ag-PEG-HER2 Nanoparticles

We have shown above that as-synthesized Ag-PEG-HER2 NPs possess both HER2-specific targeting and hyperthermic properties. To be considered as theranostic supramolecular agents, these particles should contain one more diagnostic modality. That can be easily achieved by the incorporation of the fluorescent dye into the nanoparticle structure. To this end, we modified these Ag-PEG-HER2 particles with a fluorescent DyLight650 label with fluorescence excitation and emission spectra coming through the tissue transparency NIR I window (excitation maximum is 652 nm, and emission maximum is 672 nm). The fluorescent modification is described in detail in the Materials and Methods section.

SKOV3-1ip cells were labeled with fluorescently modified Ag-PEG-HER2 NPs and visualized by two-channel confocal microscopy. The cells′ nuclei were labeled with Hoehst33342 for fluorescent cell identification. Data presented in Figure 5a demonstrate that after a specific binding to the cell surface membrane, these particles readily internalized into cells within 2 h of incubation.

The excellent imaging capabilities of such fluorescent silver particles were confirmed with flow cytometry in comparison with the anti-HER2 IgG antibody, Trastuzumab. Specifically, we used a panel of cell lines possessing a different level of HER2 expression: (i) overexpression—SKOV3-1ip, BT474, and SK-BR-3 cells, (ii) normal expression—A549 and MCF-7 cells, and (iii) no expression—CHO cells [42]. We found that while Ag-PEG-HER2 particles outperform IgG in terms of median fluorescence intensity for HER2-overexpressing cell targeting (for HER2-overexpressing cells, the MFI of cells labeled with NPs is higher than for cells labeled with IgG), the labeling of cells with a normal HER2 level is much lower than that of the antibody IgG (Figure 5b).

This fact is probably explained by the greater avidity of Ag-PEG-HER2 nanoparticles to the cell surface than to a molecular antibody. Since several small functionally active affibody molecules are presented on the surface of the nanoparticles, such a nanoparticle acts as an ensemble of recognizing molecules, exhibiting a high binding efficiency when there are many HER2 molecules on the cell surface (e.g., SKOV3-1ip, BT474, and SK-BR-3 cells) and not showing such efficiency when there are few HER2 molecules (A549 and MCF-7 cells). Such nanoparticle-assisted multivalent binding provides the improved specificity of nanoparticles in comparison with one molecule of anti-HER2 IgG.

This property of synthesized targeted nanoparticles is very promising for in vivo applications implying the reduction of side effects caused by NPs binding to normal cells while retaining the targeting efficiency for HER2-overexpressing cells.

### 3.5. Cytotoxic Properties of Ag-PEG-HER2 Nanoparticles

As-obtained HER2-directed Ag-PEG-HER2 nanoparticles were used for the targeted delivery to HER2-overexpressing cells and light-induced hyperthermia leading to specific cell death. Cells were incubated with Ag-PEG-HER2 nanoparticles, washed from non-bound particles, seeded into a 96-well plate, and irradiated with a blue LED matrix. A total of 48 h later, the cytotoxicity test, namely the MTT assay, was performed. The MTT assay results presented in Figure 6a show that the cytotoxicity of Ag-PEG-HER2 nanoparticles increased when their concentration increases, and the irradiation of cells led to cell death in a nanoparticle concentration-dependent manner.

The cytotoxicity is caused most probably by the nanoparticle degradation and silver ion Ag^+^ release leading to the generation of reactive oxygen species, ROS [43]. The ROS generation measurement upon the blue light irradiation of cells labeled with affibody-decorated Ag NP was carried out using the general oxidative stress indicator CM-H_2_DCFDA. The fluorescence intensity of cells labeled with the ROS indicator and incubated with targeted nanoparticles followed by light irradiation was measured by flow cytometry. The fluorescence intensity of cells was found to be dependent on the nanoparticle concentration and irradiation time as presented in Figure 6b. The most intensive ROS generation was observed in the cell sample incubated with 50 μg/mL NPs and irradiated for 5 min and was equal to 74.6 × 10^3^ r.u. exceeding the fluorescence of the control cell sample (without NPs and without irradiation) 4.4 times.

Since the MTT assay results sometimes do not reflect the cytotoxicity of compounds in terms of in vivo toxicity, we performed an additional test—clonogenic assay—reflecting the proliferative activity of cells and their ability to form a colony from a single cell. The clonogenic assay results demonstrate a decrease in the SKOV3-1ip cells′ proliferative activity after exposure to targeted nanoparticles under light irradiation (Figure 6c). The average number of cell colonies decreased with an increase in the Ag-PEG-HER2 concentration in the cell culture medium. The highest tested concentration of NPs (17.6 μg/mL) had the greatest cytotoxic effect: the number of colonies was 2.4 times smaller for cells without irradiation compared to the control sample (without NPs and without irradiation), 2.8 times smaller when cells were irradiated immediately, and 3.4 times smaller when cells were irradiated after 120 min after the incubation with nanoparticles (Appendix A).

### 3.6. Photothermal Cancer Therapy with Targeted Silver Nanoparticles and External Irradiation on the Xenograft Mouse Model

The therapeutic efficiency of the developed targeted Ag-based photothermal sensitizers was studied using the mouse xenograft tumor model with HER2 overexpression. This model is based on human mammary gland carcinoma BT474 cells that stably express luciferase NanoLuc in the cytoplasm and overexpress the HER2 oncomarker on their surface; these cells were obtained and thoroughly characterized by us previously [32,44,45,46,47]. We have shown that when this xenograft tumor model is injected into the right flank of the mouse, it is capable of producing metastasis on the opposite left side in the inguinal lymph node [32].

To obtain the solid tumors, BT/NanoLuc cells in Matrigel were subcutaneously injected into the right flank of mice. The overexpression of receptor HER2 in the xenograft tumors was confirmed with the immunohistochemistry assay with the HercepTest; the primary tumor tissue and respective control samples are presented in Figure 7a confirming the retention of HER2 expression on BT/NanoLuc cells in vivo.

Mice were divided into three groups and treated with injections of (i) group 1—Ag-PEG-HER2 nanoparticles followed with light irradiation, (ii) group 2—Ag-PEG-HER2 nanoparticles, and (iii) group 3—PBS (control group). All experimental groups were investigated in terms of tumor volume measurement (Figure 7b), toxicity study (Appendix A), and bioluminescent imaging (Figure 7c).

The primary tumor volume was measured with the caliper (Figure 7b). Photothermal therapy with targeted silver Ag-PEG-HER2 NPs and external irradiation of BT/NanoLuc tumors leads to full cancer remission. The smallest average tumor size was observed under the treatment with targeted NP injections in combination with light irradiation. Starting from the 57th day of the experiment, tumor elimination was observed, which indicates complete remission for at least 33 days.

A histology study (Appendix A) revealed that the liver, spleen, and heart did not exhibit significant differences in all experimental groups. However, in the lungs of mice from groups 1 and 2 treated with Ag NPs, dilated blood vessels were observed, and the grade of dilatation was comparable in both groups. In group 2, a slight expansion of the peribronchovascular space with eosinophilic edema was detected. In the kidney of mice from group 2, a significant hemorrhage and necrosis foci were found in the cortex and medulla, and irregular blood filling was observed in the medulla blood vessels. Degenerative lesions of kidneys were also detected in group 1, but they were less pronounced (small hemorrhage foci in the cortex were observed). Together with histology, various blood biochemical parameters that are indicative of organ toxicity were studied. Liver function was determined by serum activity of alanine aminotransferase (ALT) and aspartate aminotransferase (AST). Nephrotoxicity was assessed by creatinine (CREA) and urea concentrations. Cardiac damage was assessed by lactate dehydrogenase (LDH) activity. Hepatobiliary or bone damage was assessed by alkaline phosphatase (ALP) activity (Appendix A). The results of the analysis (Appendix A) demonstrate that in the long-term period, biochemical parameters did not differ significantly in all three experimental groups, which, together with the results of histology, reveal the moderate toxicity of the tested nanoparticles.

Despite the described toxicity of Ag NPs, the bioluminescent imaging studies (Figure 7c) have shown that therapy with targeted nanoparticles Ag-PEG-HER2 together with light irradiation makes it possible to completely eliminate metastases. The data presented in Figure 7 demonstrate that treatment with targeted Ag NPs without irradiation leads to the significant reduction of the primary tumor size. However, the in vivo bioimaging study showed that treatment with Ag-PEG-HER2 followed by light irradiation leads not only to the complete elimination of the primary tumor but also to the prevention of metastatic spread (Figure 7c).

## 4. Discussion

Photothermal therapy is based on the activation of photosensitizing agents by electromagnetic radiation and heat generation upon such irradiation. Nanoparticles of different nature, including LSPR-possessing ones, are among the most promising sensitizers for photothermal cancer therapy [48,49]. Localized surface plasmon resonance, LSPR, is the resonant oscillations of electrons when a surface plasmon is excited at its resonant frequency by an external electromagnetic wave [50,51]. The absorbed external electromagnetic radiation can be further dissipated by LSPR nanoparticles with the release of thermal energy, which leads to significant heating of the microenvironment. Thus, the LSPR phenomenon is one of the main tools for the development of laser-induced hyperthermia of cancer tissue labeled with nanoparticles.

To date, a wide range of plasmonic nanomaterials has been discovered, including silver, gold, platinum, copper, titanium nitride, zirconium nitride nanoparticles, and many other hybrid structures. In particular, silver nanoparticles were extensively used for plasmonic nanolithography [52], near-field scanning optical microscopy [53], surface-enhanced Raman scattering [54], catalyst particles [55], and other biophysical and biosensor applications.

Although silver is considered to be a potentially superior plasmonic nanomaterial [56,57,58], its application for cancer hyperthermia has been previously underestimated.

Different plasmon-resonant nanoparticles were used for in vivo particle tracking and photothermal therapy, e.g., in vivo tracking of gold nanostars [59] or spherical gold nanoparticles for phototherapy [60]. Gold nanospheres [61,62], nanoshells [63], and nanorods [47,64] were used as multimodal agents for biomedical applications. Full cancer remission was achieved with gold nanoshell-mediated hyperthermia by NIR irradiation [65]. As for silver particles, previous studies directed to the development of in vivo effective Ag NPs resulted in nontargeted nanoparticles or peptide-modified particles that were used for in vivo imaging [66] and tumor growth reduction [67,68,69,70,71,72,73]. In particular, TAT peptide-coated Ag NPs were used to inhibit B16 melanoma growth [67,74]; PVP-stabilized Ag NPs were developed for triple-negative breast cancer treatment [73]; and mouse serum albumin-coated Ag NPs were designed for the reduction of murine fibrosarcoma growth [69]. However, to the best of our knowledge, no targeted silver nanoparticles were created for tumor growth reduction until today.

Here, for the first time, we describe the application of plasmonic silver nanoparticles for targeted photothermal cancer therapy. This became possible thanks to an efficient method for the synthesis of nanoparticles using green chemistry and their specific modification by recognizing polypeptide molecules called affibodies.

At present, various physicochemical processes are widely used for the synthesis of NPs, which make it possible to obtain particles with the desired characteristics. However, these production methods are usually expensive, labor-intensive, and potentially hazardous to the environment and living organisms. Plant metabolites are especially promising for green synthesis: the low cost of growing plants, relatively short production times, and their safety and scalability make plants an attractive platform for the nanobiosynthesis [75,76,77,78].

It has been shown that plant metabolites such as sugars, terpenoids, polyphenols, alkaloids, phenolic acids, and proteins play an important role in the reduction of metal ions during the synthesis of NPs and in maintaining their stability. Many members of the Lamiaceae Mart. family, such as mint, basil, thyme, sage, lavender, etc., are valuable medicinal and essential oil plants, the main secondary metabolites of which are terpenoids and flavonoids. Such plants are very promising objects for research in the field of the green synthesis of metal NPs. Silver and gold NPs were synthesized with the aqueous extract of *Lavandula angustifolia* Mill. [79,80]. Studies of the toxicity of such NPs have shown that NPs are nontoxic at a wide concentration range and have more active antimicrobial and antioxidant properties than extracts. However, no studies have been performed on the creation of NPs from lavender plants reproducibly grown in in vitro culture, and their hyperthermic and antitumor properties have not been studied in xenograft tumor models.

For the effective application of the synthesized nanoparticles for cancer treatment, it is necessary to significantly reduce the side effects and accumulation of nanoparticles in the organs of the macrophage phagocytic system. Monoclonal antibodies and their derivatives have historically been widely used for targeted nanoparticle delivery to specific cells via the selective recognition of cancer cell surface receptors.

However, the use of synthetic nonimmunoglobulin-recognizing scaffold proteins, such as affibodies, DARPins, knottins, adnectins, and adhirons, gives rise to the development of new-generation targeted anticancer therapy methods and avoids a spectrum of undesirable effects [81]. These unwanted effects relate both to the design of nanoagents with full-size IgG, such as difficulty in oriented chemical modification and large-scale biotechnological production in eukaryotes with post-translational modifications, and to the nanoagents’ behavior in the organism, such as undesirable immunomodulation and rapid clearance from the bloodstream. Small scaffold proteins, namely affibodies (8 kDa), which are obtained by phage, cellular, or ribosomal display technologies, appear to be much more effective tools for targeting nanostructures to cancer cells than traditional IgGs.

The prospect of using affibodies as a tool for diagnostics and cancer treatment is confirmed by a wide range of fundamental research directed toward the design of diagnostic and therapeutic structures [82,83] and ongoing clinical trials of a radioactively labeled anti-HER2 affibody Z_HER2:342_ for the treatment of human breast cancer [29]. Thus, the ease of large-scale biotechnological production in bacteria and the absence of immunogenicity in vivo make affibodies one of the most attractive molecules for the targeted delivery of silver nanoparticles to tumor cells in vivo.

In this study, we present Ag nanoparticle-based targeted sensitizers of photothermal therapy, which is a combination of silver NPs and the anti-HER2 affibody Z_HER2:342_. As the first important result, NPs showed optimal monodispersity and size for further research due to defined synthesis conditions. The size of NPs is of great importance for in vivo applications since it determines NP properties and affects their penetration through biological barriers [84]. Therefore, it is necessary to take into consideration all parameters that affect the particle size and control these parameters to obtain optimal and successful cell penetration NPs.

Next, we demonstrate that successful modification of Ag NPs with anti-HER2 affibody Z_HER2:342_ resulted in Ag-PEG-HER2 nanoparticles that selectively interact only with HER2-overexpressing cancer cells. HER2, human epidermal growth factor receptor 2, is a clinically relevant oncomarker that is overexpressed in 20–30% of human breast cancer, ovarian cancer, and prostate and kidney carcinoma. HER2 in some subtypes of breast cancer leads to increased proliferation and angiogenesis and dysregulation of apoptosis, and it is also associated with poor prognosis, a high risk of metastasis, and reduced overall survival. Thus, the targeted therapy for HER2-positive cancer is of great social importance [85].

The as-obtained targeted Ag-PEG-HER2 NPs possessing the LSPR phenomenon are preferential sensitizers of cancer cells to local hyperthermia (application of temperatures within the 42–47 °C range). The LSPR peak of NPs is formed upon the exposure of incident light of 430–480 nm, which corresponds to visible blue light. Based on these facts, the hyperthermal properties of Ag NPs were studied upon excitation with blue light with a maximum wavelength of 465 nm. We showed that irradiation allows crossing the temperature threshold at which cancer cells are damaged but healthy human cells are not. In addition, it is important to note that NPs exhibit a wide range of temperatures depending on NP concentration and irradiation power, which will allow varying these conditions for more effective therapy of cancers of different origin. Of course, it is important to note that blue light irradiation is not the best option for in vivo therapy due to the low light penetration depth into biological tissues. However, we believe that further development of this research area using two-photon excitation by picosecond lasers, which leads to the heating of deep-seated tissues doped with nanoparticles, will give rise to the development of Ag-based cancer PTT methods.

Based on the MTT assay and clonogenic analysis results, we showed that the cytotoxic activity of Ag-PEG-HER2 NPs is significantly enhanced when cells with NPs are irradiated with blue light at moderate power. It should be noted that the clonogenic assay revealed that Ag NPs have successfully shown themselves as a cytotoxic agent that reduces the proliferative activity of cells; thus, Ag NPs can potentially be considered as anticancer agents even without the use of external light radiation.

As the main result of this study, we demonstrated complete cancer remission on xenograft tumors with HER2 overexpression. The 100% tumor growth inhibition was observed in the experimental group treated with the injections of Ag-PEG-HER2 NPs with further external light irradiation of the tumor site. Moreover, such a treatment option eliminated not only the primary tumors but also completely prevented the spread of metastatic cells. Based on these tumor therapy results, we believe that targeted LSPR-possessing silver nanoparticles is a next-generation drug for the treatment of aggressive metastatic breast cancer. Of course, before considering further studies of the efficacy of such targeted PTT, it is necessary to resolve a number of issues related to the systemic toxicity of injected substances and focus on the reduction of the doses of administered compounds. This can be done by the prolongation of the circulation time of nanoparticles in the bloodstream [86,87] or by improving the specificity of NPs by dual targeting via different oncomarkers [32,44]. 

## 5. Conclusions

This work is a proof-of-concept study focusing on Ag NP-based sensitizers for photothermal therapy that leads to 100% tumor growth inhibition and complete prevention of metastatic spread. We synthesized targeted agents for the cancer PTT, which is a combination of Ag NPs and anti-HER2 affibody Z_HER2:342_. The LSPR property of Ag NPs enhances the therapeutic effect by heating targeted NPs inside the HER2-positive cells. Ag NPs mediate the transformation of light energy into thermal energy, which leads to local hyperthermia and cancer cell death both in vitro and in vivo.

The one-pot green synthesis of Ag NPs and effective biotechnological production of the HER2-targeting agent, affibody Z_HER2:342_, allows us to consider the designed targeted structures as promising agents for rapid translation into clinical practice.

## Figures and Tables

**Figure 1 pharmaceutics-14-01013-f001:**
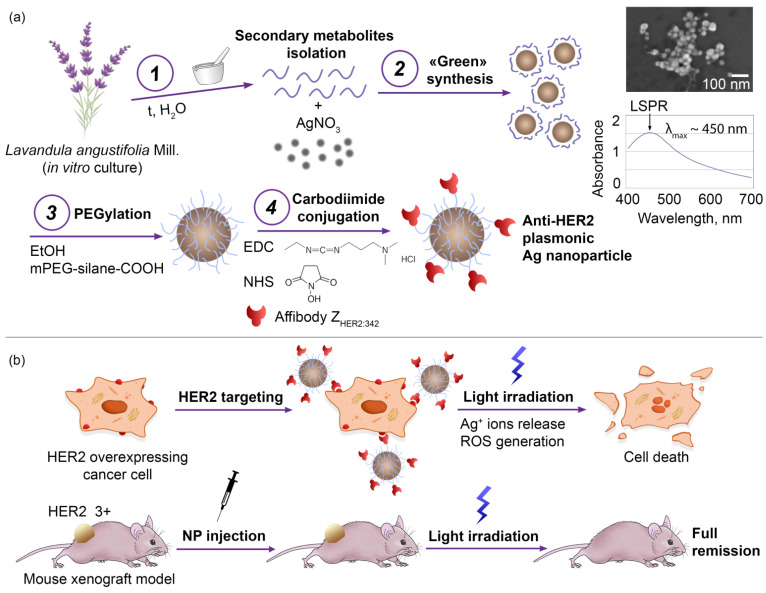
Experimental design of Ag NP synthesis and modification for the HER2-targeted photothermal therapy in vitro and in vivo. (**a**): “Green” nanoparticle synthesis via the reduction of silver nitrate with *Lavandula angustifolia* Mill. secondary metabolites (1–2), modification of NP surface with scaffold polypeptide—affibody Z_HER2:342_ through the intermediate PEGylation (3), and carbodiimide conjugation (4). Electron micrographs of Ag NPs, obtained by SEM (scale 100 nm). Absorbance spectra of Ag NPs indicate the presence of an LSPR peak. (**b**): Photothermally-induced death of HER2-overexpressing cancer cells in vitro mediated by the targeted delivery of modified NPs to specific cells and subsequent light irradiation, leading to Ag NP heating, Ag^+^ ions release, and ROS generation. in vivo xenograft tumor photothermal therapy with targeted Ag NP injections and external irradiation of the tumor site, leading to the full tumor and metastasis elimination.

**Figure 2 pharmaceutics-14-01013-f002:**
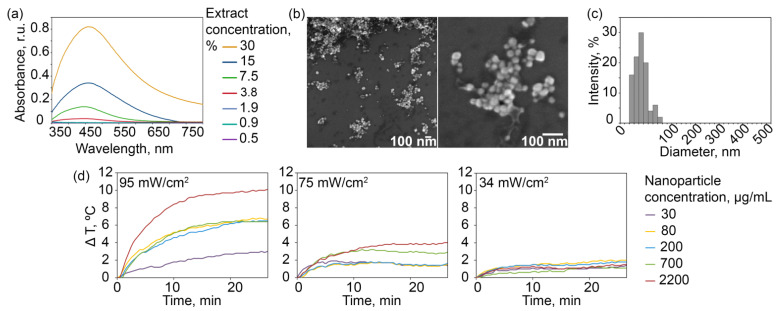
Synthesis, characterization, and optical and photothermal properties of Ag NPs. (**a**): Absorption spectra of NPs in the wavelength range of 350–800 nm, obtained by spectrophotometry during the 240 min incubation of an aqueous solution of silver nitrate (1 g/L) and plant extract at 0.5–30% concentrations. (**b**): Electron micrographs of Ag NPs obtained by scanning electron microscopy (scales, 100 nm). (**c**): Histograms of the distribution of NP number (%) by size (nm). (**d**): Dynamics of temperature change (°C) of different NP concentrations (g/L) under the irradiation with a LED matrix at a power of 95, 75, and 34 mW/cm^2^.

**Figure 3 pharmaceutics-14-01013-f003:**
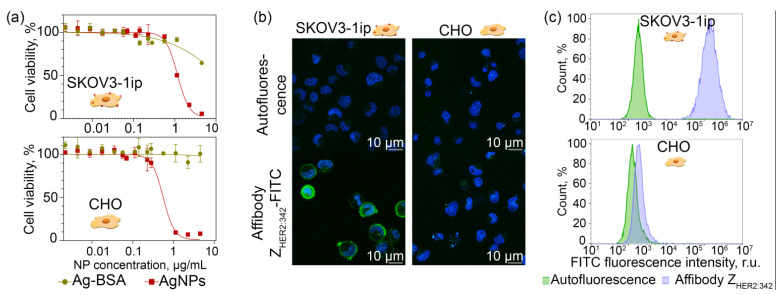
Protein-mediated shielding of Ag NP cytotoxicity. Affibody Z_HER2:342_ cell labeling. (**a**): MTT assay results of the SKOV3-1ip and CHO cells viability (%) depending on the concentration of unmodified Ag NPs and BSA-modified Ag-BSA NPs in the medium (µg/mL). (**b**): Confocal microscopy of HER2-overexpressing SKOV3-1ip and HER2-negative CHO cells labeled with Z_HER2:342_-FITC. (**c**): Flow cytometry histograms of SKOV3-1ip and CHO cells showing the cell interaction with Z_HER2:342_-FITC. Green—cell autofluorescence, blue—cells labeled with affibody.

**Figure 4 pharmaceutics-14-01013-f004:**
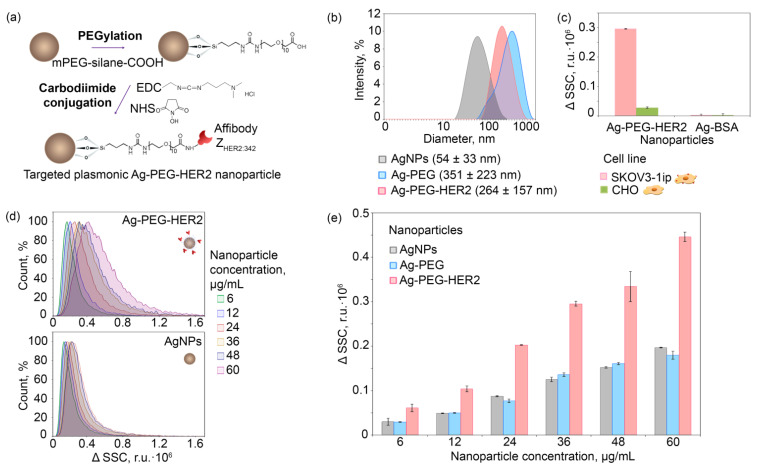
Nanoparticle conjugation to HER2-recognizing scaffold protein Z_HER2:342_ and HER2-overexpressing cell targeting in in vitro tests. (**a**): The schematic illustration of NP surface coating with artificial scaffold polypeptide—affibody Z_HER2:342_ through the intermediate PEGylation and subsequent carbodiimide conjugation. (**b**): DLS distributions showing the dependence of the intensity (%) of pristine (Ag NPs), PEGylated (Ag-PEG), and affibody-decorated (Ag-PEG-HER2) silver nanoparticles on their hydrodynamic diameter (nm). (**c**): Flow cytometry side scatter change ΔSSC of SKOV3-1ip and CHO cell populations labeled with targeted Ag-PEG-HER2 and nontargeted Ag-BSA nanoparticles (r.u.·10^5^). (**d**): Flow cytometry histograms in SSC channel of SKOV3-1ip cells labeled with different concentrations of Ag NPs (top) and Ag-PEG-HER2 (bottom) nanoparticles. (**e**) The dependence of ΔSSC (r.u.·10^3^) on SKOV3-1ip cell populations labeled with different concentrations of Ag-PEG-HER2, Ag-PEG, and Ag (µg/mL) obtained with the flow cytometry. ΔSSC was calculated as raw SSC intensity with subtracted SSC intensity of unstained cell samples.

**Figure 5 pharmaceutics-14-01013-f005:**
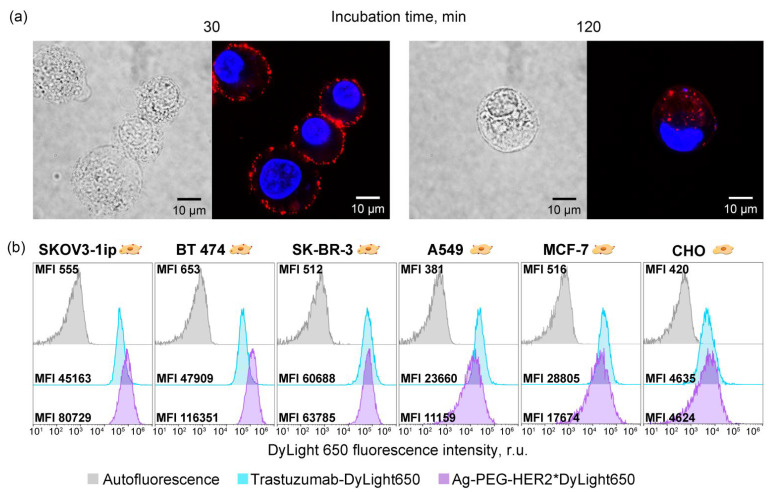
Imaging modalities of Ag-PEG-HER2 nanoparticles. (**a**): Visualization of the internalization of Ag-PEG-Z_HER2:342_ NPs in SKOV3-1ip cells after incubation for 30 (left panels) and 120 (right panels) min using two-channel confocal microscopy. Cell nuclei are stained with Hoechst33342 (blue), and NPs are labeled with DyLight650 (red). (**b**): Flow cytometry histograms of SKOV3-1ip, BT474, SK-BR-3, A549, MCF-7, and CHO cells showing the cell interaction with Trastazumab-DyLight650 and Ag-PEG-HER2*DyLight650. Gray—cell autofluorescence; blue—cells labeled with Trastuzumab; and violet—cells labeled with Ag-PEG-HER2*DyLight650 nanoparticles.

**Figure 6 pharmaceutics-14-01013-f006:**
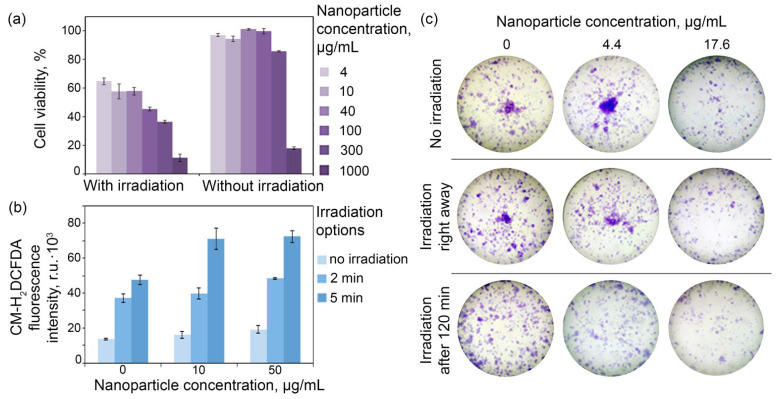
Light-induced cytotoxicity of Ag-PEG-HER2 nanoparticles in vitro. (**a**): Viability of SKOV3-1ip cells (%) depending on the concentration of targeted silver NPs in the medium (μg/mL) (shown in different colors) and the irradiation mode (with/without light) obtained with the MTT test. (**b**): ROS generation upon the blue light irradiation of cells labeled with affibody-decorated Ag NPs: dependence of the fluorescence intensity (r.u.·10^3^) of cells labeled with ROS indicator on the concentration of nanoparticles and irradiation time. (**c**): Colony formation assay showing the cytotoxic activity of combination affibody-modified Ag-PEG-HER2 particles and irradiation with the blue LED matrix at a power of 95 mW/cm^2^ for HER2-positive cells.

**Figure 7 pharmaceutics-14-01013-f007:**
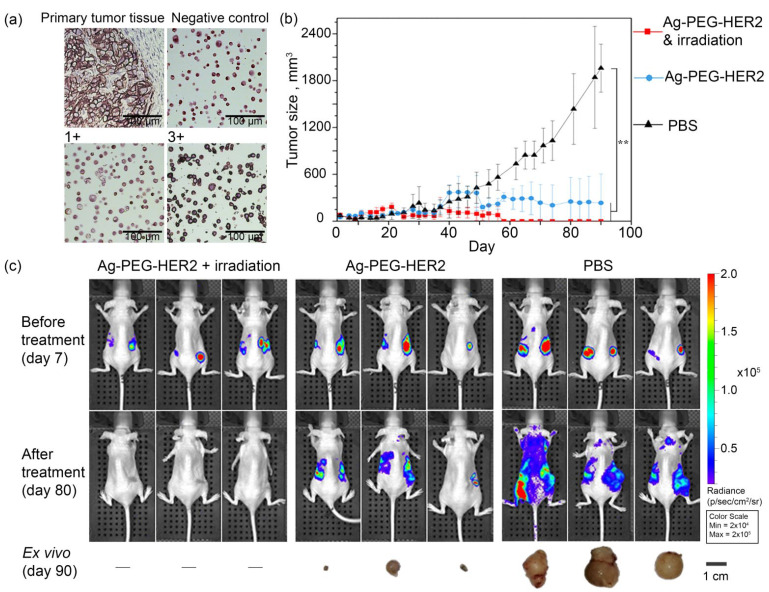
PTT of xenograft tumors with targeted Ag-PEG-HER2 nanoparticles and light irradiation. (**a**): Immunohistochemistry of BT/NanoLuc primary tumor tissue and HercepTest controls: negative control (0), normal HER2 expression (1+), and HER2 overexpression (3+) (scale 100 µm). (**b**): Dynamics of BT/NanoLuc tumor growth under the treatment with repeated injections of PBS, Ag-PEG-HER2 nanoparticles, and Ag-PEG-HER2 with 1 h post blue light irradiation (n = 3 for each group), ** *p* < 0.05. (**c**): Bioluminescent imaging of BT/NanoLuc xenograft tumors before treatment (day 7 after inoculation) and after treatment (day 80) and ex vivo imaging of primary tumors (day 90).

## Data Availability

All data are presented within the manuscript and Appendix A.

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
