# Peer review of "Photothermal Therapy with HER2-Targeted Silver Nanoparticles Leading to Cancer Remission"

_pharmaceutics, 2022, doi:10.3390/pharmaceutics14051013_

Round 1

Reviewer 1 Report

It’s interesting to see the plasmonic Ag nanoparticles are proved as effective sensitizers for targeted photothermal oncotherapy for the first time. The experimental design and validation look concise and feasible. However, I would like to ask authors to address minor questions as follows:

  1. Page 3, line 98. “Materials” section seems missing. The “Materials and Methods” only contains “Methods”. Please add detailed information about materials.
  2. Page 4, line 177. “2.6 Cell culture” section should clearly state the species and source of all cell cultures used in experiments.
  3. Page 6, line 278 and 293. 2.14 and 2.15 shared the same title. Please confirm the titles of the two sections.
  4. Page 7 line 312. The “3.1 Experimental design” might be more suitable in “Methods” section rather than “Results” section.
  5. Page 10. Figure 3(a), the abscissa axis only showed the NP concentrations from 0.01 to 1.0 µg/mL, but the script says IC 50 for CHO cells was 116.4 µg/mL. Please confirm the figure was correctly presented. And could authors provide references for why they choose IC 50 for the standard of cytotoxicity because normally the cell viability needs to be higher to prove a substance is non-cytotoxicity (Xing et al., 2015).

Xing, J., Wang, G., Zhang, Q., Liu, X., Yin, B., Fang, D., Zhao, J., Zhang, H., Chen, Y. Q., & Chen, W. (2015). Determining antioxidant activities of lactobacilli by cellular antioxidant assay in mammal cells. Journal of Functional Foods, 19, 554–562. https://doi.org/10.1016/j.jff.2015.09.017

Author Response

We sincerely thank the reviewer for the time and effort spent on reviewing the manuscript. All comments were thoroughly addressed and corrections were incorporated to the manuscript as follows:

Comment 1

Page 3, line 98. “Materials” section seems missing. The “Materials and Methods” only contains “Methods”. Please add detailed information about materials.

Reply 1

The concern is reasonable. The “Materials” section was added to the “Materials and Methods”.

Comment 2

Page 4, line 177. “2.6 Cell culture” section should clearly state the species and source of all cell cultures used in experiments.

Reply 2

Corrected. The corresponding section was revised as follows:

“Human breast adenocarcinoma BT-474 (ATCC HTB-20), SK-BR-3 (ATCC HTB-30) and MCF-7 (ATCC HTB-22) cells, human ovarian adenocarcinoma SK-OV-3-1ip cells (Shemyakin-Ovchinnikov Institute RAS, Molecular Immunology Laboratory collection, [33]), human lung carcinoma A549 (ATCC CCL-185) cells, and Chinese hamster ovary CHO (Shemyakin-Ovchinnikov Institute RAS, Molecular Immunology Laboratory collection) cells were cultured in DMEM medium (HyClone, Logan, UT, USA) supplemented with 10% fetal bovine serum (HyClone, Logan, UT, USA), penicillin/streptomycin (PanEko, Moscow, Russia) and 2 mM L-glutamine (PanEko, Moscow, Russia). Cells were incubated under a humidified atmosphere with 5% CO2 at 37 °C. BT/NanoLuc cells were obtained by us previously and used without any modifications [32]”.

Comment 3

Page 6, line 278 and 293. 2.14 and 2.15 shared the same title. Please confirm the titles of the two sections.

Reply 3

The titles of sections were corrected.

Comment 4

Page 7 line 312. The “3.1 Experimental design” might be more suitable in “Methods” section rather than “Results” section.

Reply 4

Corrected, the “Experimental Design” was transferred to “Materials and Methods” section.

Comment 5

Page 10. Figure 3(a), the abscissa axis only showed the NP concentrations from 0.01 to 1.0 µg/mL, but the script says IC 50 for CHO cells was 116.4 µg/mL. Please confirm the figure was correctly presented. And could authors provide references for why they choose IC 50 for the standard of cytotoxicity because normally the cell viability needs to be higher to prove a substance is non-cytotoxicity (Xing et al., 2015).

Reply 5

The abscissa axis was corrected, namely, the concentration range used in the experiment started from 4.5 µg/mL. The IC50 equal to 116.4 µg/mL is calculated using the approximation of the cytotoxicity curve with “[Inhibitor] vs. response” = “variable slope (four parameters)”. Thus, the 116.4 µg/mL is just the point on the extrapolation of the cytotoxicity curve. The IC50 calculated in this work is a relative IC50, not absolute, which means that it is the concentration required to bring the curve down to point half way between the top and bottom plateaus of the curve.

However, it can be observed from the data presented in Figure 3a that the concentrations of Ag-BSA smaller than 2 µg/mL leads to the cell viability of at least 85% for SKOV3-1ip cells and 90 % for CHO cells. This fact allows us to consider these particles as non-toxic under such treatment conditions. This fact was reflected in the text as follows:

“Indeed, MTT assay results show that cell viability significantly increased after protein shielding of the NP surface, namely IC50 equal to 1.17 vs. 11.6 µg/mL for unmodified and BSA-coated particles, respectively, for SKOV3-1ip cells, and IC50 equal to 0.55 vs. 116.4 µg/mL for CHO cells. We should note that IC50 calculated here is a relative IC50 (not absolute IC50), which means that it is the concentration required to bring the curve down to point half way between the top and bottom plateaus of the curve. Thus, taking into account the facts that coating of nanoparticles with BSA leads to significant increase of IC50 and the cell viability is at least 85% for SKOV3-1ip cells and 90 % for CHO cells at concentrations from 2 µg/mL, such particles can be considered as non-toxic under such treatment conditions [37]”.

Reviewer 2 Report

It was a good study about the green synthesis of Ag nanoparticles functionalized with affibody for the photothermal therapy of cancer cells with overexpression of HER2. Here are some comments on this study that should be considered before publication:

1- There are some typos and grammatical mistakes in the text that should be corrected. Some of them are listed as follows:

  • "tracts in the concentration range of 0.5 ÷ 30%"
  • " containing 100 μg/mL kanamycin at 25 °C for 24 h at 25 °C."
  • "Mice were s.c. injected with 2·106"
  • “While tumor treatment options, with Ag-PEG-HER2 with or without irradiation, do not significantly affect tumor size”

2- Please add a material subheading and describe the information of your materials in it. 

3- "After 5 min of incubation at room temperature, the protein was mixed with NPs at 1.8 g/L in a 1.75:1 v/v ratio, the suspension was gently mixed, briefly exposed to ultrasound, and incubated for 8 h at room temperature. Next, phosphate-buffered saline (PBS) with 1% of BSA was added to the mixture of Ag NPs" did you add BSA 2 times? Or here you have mistakes?

4- Did you use physical conjugation for the attachment of PEG to Ag NPs? 

5- Please add more analytical tests for confirming the synthesis and modification of the nanoparticle.

6- Which cell line(s) did you use for the cell culture study? you should mention it in section 2.6.

7- For which purpose did you use "Trastuzumab"? Please mention it in section 2.7.

8- Why did you prepare Ag-BSA particles? Did you add PEG to these particles or Ag alone?

9- Why did you use different cell lines just for the flow cytometry test?

10- Please add the information of the LED matrix used in section 2.10.

 11- Please test the effect of nanoparticles without irradiation in sections 2.11 and 2.12.

12- "Mice were s.c. injected with 2·106" what does "s.c." refer to?

13- Why size of the particles decreased after the addition of the targeting agent?

14- “We found that while Ag-PEG-HER2 particles outperform IgG in terms of median fluorescence intensity for HER2-overexpressing cells targeting (for HER2-overexpressing cells MFI of cells labeled with NPs is higher than for cells labeled with IgG), the labeling of cells with normal HER2 level is much lower than that of antibody IgG” why this happened? What is your explanation for this event?

15- There are differences between the time of irradiation in the “ROS Generation Study” in the method and results parts!

16- Based on the results shown in figure S1, nanoparticles without irradiation also showed toxicity effects even in low concentrations! How do you explain this?

17- Please compare the results of your study with other similar research in the discussion section.

18- It is not normal to use references in the conclusion section.

Author Response

We are grateful to the reviewer for the time and effort spent on reviewing the manuscript. Additional experiments were performed and all issues were addressed as follows:

Comment 1

There are some typos and grammatical mistakes in the text that should be corrected. Some of them are listed as follows:

"tracts in the concentration range of 0.5 ÷ 30%"

" containing 100 μg/mL kanamycin at 25 °C for 24 h at 25 °C."

"Mice were s.c. injected with 2·106"

“While tumor treatment options, with Ag-PEG-HER2 with or without irradiation, do not significantly affect tumor size”

Reply 1

The typos and grammatical mistakes were corrected.

Comment 2

Please add a material subheading and describe the information of your materials in it.

Reply 2

The concern is reasonable. The “Materials” section was added to the “Materials and Methods”.

Comment 3

"After 5 min of incubation at room temperature, the protein was mixed with NPs at 1.8 g/L in a 1.75:1 v/v ratio, the suspension was gently mixed, briefly exposed to ultrasound, and incubated for 8 h at room temperature. Next, phosphate-buffered saline (PBS) with 1% of BSA was added to the mixture of Ag NPs" did you add BSA 2 times? Or here you have mistakes?

Reply 3

This is a mistake which was corrected to “Next, phosphate-buffered saline (PBS) was added to the mixture of Ag NPs”.

Comment 4

Did you use physical conjugation for the attachment of PEG to Ag NPs?

Reply 4

To attach PEG to Ag NPs we used chemical conjugation using the silane chemistry, namely between hydroxyl group of Ag NP surface and methoxyl silane functionalized polyethylene glycol (mPEG-silane-COOH), which is described in detail in “2.7. Nanoparticle Modification: BSA Stabilization, PEG Modification, Affibody Conjugation” section.

Comment 5

Please add more analytical tests for confirming the synthesis and modification of the nanoparticle.

Reply 5

Lateral flow assays were performed to confirm the modification of nanoparticles during all steps of the conjugation. The respective results are presented as follows:

Supporting Note 1

Methods

Affibody labeled with biotin was prepared as follows. 130 µL of affibody ZHER2:342 in PBS at 1 g/L was mixed with 10 µL of biotin-NHS ester (Thermo Fisher, USA, EZ-Link NHS-Biotin) in DMSO at 8 g/L and incubated for 2 h at room temperature. The excess of unreacted molecules was removed with Zeba Spin Desalting Columns 7 kDa MWCO (Thermo Fisher, USA) according to the manufacturer’s recommendations. The concentration of protein was determined with the BCA protein assay. The conjugation of biotin-labeled affibody with Ag-PEG particles was performed absolutely in the same way as for pristine affibody.

For immunochromatograpy test, the 2 mm test strip (MDI, # LKDFXXX060X260X 90CNPH-N-SS40-L2-P25) was applied with 1 µL of anti-biotin IgG (IgG Fraction Monoclonal Mouse Anti-Biotin, Jackson Immunoresearch) at 1.2 g/L and air-dried. Next, the strip was placed into the tube with 40 µL Tris, 1% BSA, 0.05% Tween-20 and 2 µg of test particles. The particles were allowed to migrate until the all liquid was absorbed on absorbent pad. The strips were imaged with a smartphone camera.

Results

To validate the presence of protein – affibody ZHER2:342 on the nanoparticle surface, we performed a lateral flow assay with immunochromatograpy strips coated with anti-biotin IgG and particles conjugated to the biotinylated affibody, namely, Ag-PEG-bio-HER2.

The data presented in Fig. S1 demonstrate that uncoated Ag NPs possess slight non-spesific binding to anti-biotin IgG. However, this non-spesific binding is successfully shielded by modification of nanoparticle surface with PEG (ag-PEG). The affibody-conjugated Ag-PEG-HER2 particles do not exhibit any binding to anti-biotin IgG. And only the biotin-affibody conjugated Ag-PEG-bio-HER2 particles interact specifically with the anti-biotin IgG thus forming a strong absorbing line on the test strip thus confirming the efficiency of conjugation.

Figure S1. Immunochromatography assay on the conjugation efficiency. The anti-biotin IgG was applied to the nitrocellulose test strip. The strips were placed into the tube with: unmodified Ag NPs, PEG-modified AgNPs, AG-PEG-HER2 particles, and Ag-PEG-bio-HER2 particles (modified with biotin-labeled affibody)”.

Comment 6

Which cell line(s) did you use for the cell culture study? you should mention it in section 2.6.

Reply 6

The corresponding section was corrected as follows:

“Human breast adenocarcinoma BT-474 (ATCC HTB-20), SK-BR-3 (ATCC HTB-30) and MCF-7 (ATCC HTB-22) cells, human ovarian adenocarcinoma SK-OV-3-1ip cells (Shemyakin-Ovchinnikov Institute RAS, Molecular Immunology Laboratory collection, [33]), human lung carcinoma A549 (ATCC CCL-185) cells, and Chinese hamster ovary CHO (Shemyakin-Ovchinnikov Institute RAS, Molecular Immunology Laboratory collection) cells were cultured in DMEM medium (HyClone, Logan, UT, USA) supplemented with 10% fetal bovine serum (HyClone, Logan, UT, USA), penicillin/streptomycin (PanEko, Moscow, Russia) and 2 mM L-glutamine (PanEko, Moscow, Russia). Cells were incubated under a humidified atmosphere with 5% CO2 at 37 °C. BT/NanoLuc cells were obtained by us previously and used without any modifications [32]”.

Comment 7

For which purpose did you use "Trastuzumab"? Please mention it in section 2.7.

Reply 7

The DyLight-650 labeled Trastuzumab was used to compare imaging capabilities of Ag-PEG-HER2 particles with anti-HER2 antibody, the data are presented in the section “3.4. Imaging Modalities of Ag-PEG-HER2 Nanoparticles”.

Comment 8

Why did you prepare Ag-BSA particles? Did you add PEG to these particles or Ag alone?

Reply 8

Ag-BSA particles were prepared using the methodology described in the section “2.7. Nanoparticle Modification: BSA Stabilization, PEG Modification, Affibody Conjugation” in detail. These particles were prepared without intermediate PEG modification in order to show that cytotoxicity of Ag nanoparticles can be effectively shielded with non-specific protein that does not possess any effector or cytotoxic function. BSA was used as just a model protein in this experiment.

Comment 9

Why did you use different cell lines just for the flow cytometry test?

Reply 9

The only reason why we used a panel of six cell lines in the flow cytometry test was to show the specificity of binding of targeted Ag-PEG-HER2 nanoparticles to cells and to show that the binding of Ag-PEG-HER2 monotonically depends on HER2 expression on cell surface: the higher the expression, the higher the Ag-PEG-HER2 binding.

The data presented in Fig. 5b confirm the excellent specificity of targeted nanoparticles Ag-PEG-HER2, so it seems eligible to use only two cell lines for experiments with light irradiation and cell toxicity study.

Comment 10

Please add the information of the LED matrix used in section 2.10.

Reply 10

The information about LED matrix was added to the section as follows:

“For the photothermal-induced cytotoxicity study, after the addition of Ag NPs to cells, incubation, and washing from non-bound nanoparticles, the plate was placed un-der the 5 × 5 cm LED matrix (465 nm, power of 95 mW/cm2) and irradiated for 20 min followed by the cultivation and cytotoxicity study as described above”.

Comment 11

Please test the effect of nanoparticles without irradiation in sections 2.11 and 2.12.

Reply 11

The effect of nanoparticles without irradiation has already been studied – see Fig. 6b “no irradiation” option and Fig. 6c “no irradiation” option.

Comment 12

"Mice were s.c. injected with 2·106" what does "s.c." refer to?

Reply 12

The typo was corrected to “Mice were s.c. injected with 2·106 BT/NanoLuc cells in 100 μL of 30% Matrigel in the full culture medium”.

Comment 13

Why size of the particles decreased after the addition of the targeting agent?

Reply 13

In this experiment we measured the hydrodynamic size of nanoparticles. The increase of hydrodynamic size of Ag particles after PEG coating is explained by the fact that PEG forms a highly stable hydration shell because the spacing between adjacent ethereal oxygens matches water's hydrogen-bonding network. However, after the introduction of protein molecules into the surface of nanoparticles, such a hydrate coat destabilizes, since protein molecules destabilize this system of hydrogen bonds, making this hydrate coat smaller.

Comment 14

“We found that while Ag-PEG-HER2 particles outperform IgG in terms of median fluorescence intensity for HER2-overexpressing cells targeting (for HER2-overexpressing cells MFI of cells labeled with NPs is higher than for cells labeled with IgG), the labeling of cells with normal HER2 level is much lower than that of antibody IgG” why this happened? What is your explanation for this event?

Reply 14

The explanation was added to the main text as follows:

“This fact is probably explained by the greater avidity of Ag-PEG-HER2 nanoparti-cles to the cell surface than a molecular antibody. Since several small functionally active affibody molecules are presented on the surface of the nanoparticles, such a nanoparticle acts as an ensemble of recognizing molecules, exhibiting a high binding efficiency when there are many HER2 molecules on the cell surface (e.g. SKOV3-1ip, BT474, and SK-BR-3 cells) and not showing such efficiency when there are few HER2 molecules (A549, MCF-7 cells). Such nanoparticle-assisted multivalent binding provides the improved specificity of nanoparticles in comparison with one molecule of anti-HER2 IgG”.

Comment 15

There are differences between the time of irradiation in the “ROS Generation Study” in the method and results parts!

Reply 15

The typo was corrected to “Then samples were irradiated with LED matrix at a power of 95 mW/cm2 at 10 °C for 0, 2 and 5 min”.

Comment 16

Based on the results shown in figure S1, nanoparticles without irradiation also showed toxicity effects even in low concentrations! How do you explain this?

Reply 16

Since we show that Ag-PEG-HER2 binds specifically to HER2-positive cells within the wide concentration range and confocal microscopy study showed that particles effectively internalized into HER2-positive cells, the degradation of Ag-PEG-HER2 inside cells is an inevitable process. The degradation leads to the release of Ag+ ions which toxic to cells even at low concentrations and this is the most probable explanation of such toxicity.

Comment 17

Please compare the results of your study with other similar research in the discussion section.

Reply 17

The respective part of Discussion section was modified as follows:

“Although silver is considered to be a potentially superior plasmonic nanomaterial [56–58], its application for cancer hyperthermia has been previously underestimated.

Different plasmon-resonant nanoparticles were used for in vivo particle tracking and photothermal therapy, e.g. in vivo tracking of gold nanostars [59] or using spherical gold nanoparticles for phototherapy [60]. Gold nanospheres [61,62], nanoshells [63] and nanorods [47,64] were used as multimodal agents for biomedical applications. The full cancer remission was achieved with gold nanoshell-mediated hyperthermia by NIR irradiation [65]. As for silver particles, previous studies directed to the development of in vivo effective Ag NPs resulted in non-targeted nanoparticles or peptide-modified particles that were used for in vivo imaging [66] and tumor growth reduction [67–73]. Namely, TAT-peptide coated Ag NPs were used to inhibit B16 melanoma growth [67,74], PVP-stabilized Ag NPs were developed for the triple-negative breast cancer treatment [73] or mouse serum albumin-coated Ag NPs were designed for the reduction of murine fibrosarcoma growth [69]. However, to the best of our knowledge, no targeted silver nanoparticles were created for the tumor growth reduction until today”.

Comment 18

It is not normal to use references in the conclusion section.

Reply 18

The concern is reasonable, and the references in the conclusion were omitted.

Reviewer 3 Report

The authors green-synthesized silver nanoparticles (Ag NPs) using the aqueous extracts of Lavandula angustifolia Mill., conjugated Ag NPs to the HER2-specific affibody. They tested the photothermal therapy effects of HER2-target AgNPs in vitro and in vivo. The study is well designed and written correctly. The results are also convincing. However, the description of the concentration range of the plant extracts was not clear and consistent throughout the manuscript. Please check and revise.

Author Response

Comment 1

The authors green-synthesized silver nanoparticles (Ag NPs) using the aqueous extracts of Lavandula angustifolia Mill., conjugated Ag NPs to the HER2-specific affibody. They tested the photothermal therapy effects of HER2-target AgNPs in vitro and in vivo. The study is well designed and written correctly. The results are also convincing. However, the description of the concentration range of the plant extracts was not clear and consistent throughout the manuscript. Please check and revise.

Reply 1

The description of the concentration range of the plant extracts was corrected and is now formulated as follows:

“Silver NPs were prepared using the “green” synthesis technique by mixing 50 μL of AgNO3 solution at 1 g/L in water with the 50 μL of Lavandula angustifolia Mill. plant extracts in the concentration range starting from 30% with 2-fold serial dilution down to 0.5%”.

Round 2

Reviewer 2 Report

-